# Gut-Microbial Metabolites, Probiotics and Their Roles in Type 2 Diabetes

**DOI:** 10.3390/ijms222312846

**Published:** 2021-11-27

**Authors:** Lixiang Zhai, Jiayan Wu, Yan Y. Lam, Hiu Yee Kwan, Zhao-Xiang Bian, Hoi Leong Xavier Wong

**Affiliations:** 1School of Chinese Medicine, Hong Kong Baptist University, Kowloon, Hong Kong, China; lxzhai@hkbu.edu.hk (L.Z.); 21481679@life.hkbu.edu.hk (J.W.); hykwan@hkbu.edu.hk (H.Y.K.); 2Centre for Chinese Herbal Medicine Drug Development Limited, Hong Kong Baptist University, New Territories, Hong Kong, China; yanlam2021@hkbu.edu.hk

**Keywords:** gut microbiota, microbial metabolites, probiotics, insulin resistance, type 2 diabetes, insulin signaling

## Abstract

Type 2 diabetes (T2D) is a worldwide prevalent metabolic disorder defined by high blood glucose levels due to insulin resistance (IR) and impaired insulin secretion. Understanding the mechanism of insulin action is of great importance to the continuing development of novel therapeutic strategies for the treatment of T2D. Disturbances of gut microbiota have been widely found in T2D patients and contribute to the development of IR. In the present article, we reviewed the pathological role of gut microbial metabolites including gaseous products, branched-chain amino acids (BCAAs) products, aromatic amino acids (AAAs) products, bile acids (BA) products, choline products and bacterial toxins in regulating insulin sensitivity in T2D. Following that, we summarized probiotics-based therapeutic strategy for the treatment of T2D with a focus on modulating gut microbiota in both animal and human studies. These results indicate that gut-microbial metabolites are involved in the pathogenesis of T2D and supplementation of probiotics could be beneficial to alleviate IR in T2D via modulation of gut microbiota.

## 1. Introduction

Type 2 diabetes (T2D) is characterized by fasting hyperglycemia resulted from the inadequate secretion of the glucose-lowering hormone insulin and/or insulin resistance (IR). Primarily driven by overnutrition and sedentary lifestyles, T2D is a major global health problem in both developing and developed countries [1]. The high prevalence of IR in T2D makes IR become a predictor for the development of T2D and also an ideal therapeutic target to maintain glucose levels.

Growing evidence suggests that the gut microbiome is an important factor for the pathogenesis of IR and T2D [2]. The gut microbiota is capable of utilizing undigested and unabsorbed food components, thereby yielding bioactive metabolites from the metabolism of carbohydrate, protein, choline and primary bile acids. Many studies have pointed out that these metabolites play critical roles in the development of IR and T2D [3]. The proteolytic fermentation of gut microbiota yields products including indoles, phenols, *p*-cresol, hydrogen sulfide, branched-chain fatty acids, ammonia and polyamines. Some of them may be either beneficial or detrimental to the gut and metabolic homeostasis of the host [4]. The composition and structure of gut microbiota could be of interest to determine the effects of microbial metabolites on metabolic diseases [5]. In the present study, we review and summarize the contribution of microbial metabolites to the development of T2D, identify gaps based on the current literature, and provide a perspective on the direction of future research in this field.

Probiotics, referring to “live microorganisms which when consumed in proper amounts confer beneficial effects on the host”, has been used as a therapeutic tool for the treatment of IR and T2D [6]. Both animal and human data regarding the efficacy of the probiotics have been reported, while some of the results are contradictory. Several reasons, such as the use of probiotics strains, dosage and duration and study design, could be attributed to the differences in these studies. For this reason, we also summarized the current evidence of using probiotics as therapeutic agents for the treatment of T2D. Both animal and human data were included to address the role of probiotics in alleviating IR in T2D for further research. 

### 1.1. Gut-Microbial Metabolites and Their Roles in the Development of T2D

Microbial metabolites derived from dietary components (e.g., dietary fiber, cholesterol, amino acids) are involved in the development of metabolic diseases including IR and T2D [7]. Among dietary components, carbohydrates are fermented by microbes in the proximal colon, while the fermentation of protein mainly takes place in the distal colon; the latter occurs as the more easily digested carbohydrates are depleted, where little is known about the microbial networks that produce bile acids, choline, saccharolytic and proteolytic metabolites. Briefly, the fermentation of dietary fiber produces high amounts of short-chain fatty acids (SCFAs), lactate, succinate and gases, such as methane and carbon dioxide in the proximal colon [8]. In contrast, residual peptides and proteins, bile acids, and choline are fermented in the distal colon [9]. Compared to the fermentation of carbohydrates in the proximal colon, the fermentation products in the distal colon seems to be more diverse, including the following: (1) bacterial toxins such as lipopolysaccharides (LPS); (2) gaseous products like methane, carbon dioxide, and hydrogen sulfide; (3) bile acids (BA) products like deoxycholate and lithocholate; (4) branched-chain amino acids (BCAAs) and products like branched-chain fatty acids (BCFAs) isobutyrate, 2-methylbutyrate and isovalerate; (5) aromatic amino acids (AAAs) products like phenolic, indolic, skatolic and *p*-cresolic compounds as well as ammonia and polyamines and (6) choline products like and dimethylamine (DMA) and trimethylamine (TMA); (Figure 1). Studies have also measured the concentration of these microbial metabolites in serum, urine and feces, and the colon of humans (adults) (Table 1), which facilitate biological studies of microbial metabolites on host health.

Over the past decades, the gut microbiome has emerged as an important “organ” regulating energy metabolism in the host. Abnormalities in gut microbiota composition and function have been found to contribute to disruptions of host metabolism in T2D, including insulin-desensitizing effects on metabolism in adipose tissue, skeletal muscle and liver [20,21]. Studies have shown the gut microbiome significantly affects metabolic signatures of T2D subjects [22]. Many untargeted and targeted metabolomics studies on subjects with T2D have been reported. Although these studies were performed in different populations (Asians, Europeans and Americans) using different metabolomics approaches, they have identified several similar patterns of metabolome in T2D. First, metabolomics is useful in discriminating T2D patients from subjects with pre-diabetes and healthy subjects [23,24,25]. Secondly, numerous untargeted and targeted metabolomics studies have determined the changes of gut microbial metabolites in T2D, showing the gut microbial metabolite-related metabolic pathways are significantly changed in T2D. Both the targeted and untargeted metabolomics studies related to animals and human studies of T2D focusing on small molecules were reviewed here to address the role of microbial metabolites in IR and T2D (Figure 2).

#### 1.1.1. Bacterial Toxins and LPS

LPS is derived from the gram-negative bacterial wall and has a high binding affinity towards a series of immune-related receptors including toll-like receptors (TLRs), NOD-like receptors and the NLRP3 inflammasome which are highly expressed in macrophages and dendritic cells. LPS activates the TLR4/MyD88/NF-κB pathway to trigger inflammatory responses and the release of pro-inflammatory factors TNF-α, IL-1beta, IL-6, and iNOS. With the activation of TNF-alpha receptors, JNK and IKK are activated to downregulate the serine phosphorylation of insulin receptor substrate (IRS), which inhibits insulin signaling and results in cellular IR [26,27,28].

Patients with T2D have higher levels of blood LPS [29]. This is because intestinal permeability is increased in diabetic patients, therefore endotoxins easily penetrate the intestinal barrier [30]. Intestinal permeability is usually regulated by tight junction proteins in intestinal epithelial cells, which prevents microbes and toxins from entering the circulation. In mice with diabetes, the expressions of barrier function proteins including zonula occludens-1 (ZO-1), occludin, and claudin are reduced, leading to the translocation of bacteria and LPS into circulation [29]. Increased LPS levels may thereby contribute to the development of T2D by triggering inflammation-induced IR. In a 60-month follow-up study, postprandial LPS levels are higher in patients with T2D when compared to healthy subjects [31]. Taken together, these studies suggest increased intestinal permeability resulted from diabetes facilitates the translocation of bacteria and toxins into circulation, resulting in an increased level of LPS in serum and hence impairing glucose metabolism and insulin signaling. 

#### 1.1.2. Carbohydrate Metabolites: Short-Chain Fatty Acids (SCFAs)

Derived from undigestible foods, acetate, propionate, and butyrate, which are three major types of SCFAs, are the most abundant microbial metabolites. SCFAs are the most well-studied microbial metabolites so far and play multiple roles in IR and T2D, including promoting gut epithelial integrity, controlling immunomodulatory functions and regulating pancreatic β-cell proliferation and insulin secretion [32]. SCFAs bind to G-protein-coupled receptors 43 and 41 (GPR43/FFA2 and GPR41/FFA3) in enteroendocrine cells, intestinal epithelial cells and islets of Langerhans [33]. SCFAs stimulate the production of glucagon-like peptide (GLP-1) via FFAR2, a gut hormone that regulates glucose-dependent insulin secretion and inhibits glucagon secretion [34]. Similarly, GPR41 activation regulates intestinal gluconeogenesis and energy expenditure and stimulates intestinal peptide YY secretion in animals. Moreover, SCFAs bind to GPR119 in intestinal L-cells and pancreatic β-cells. It is shown that GPR119 agonists reduce hyperglycemia by stimulating intestinal GLP-1 secretion, improving pancreatic β-cell function and insulin secretion [35]. Therefore, SCFAs exert beneficial properties via activation of G-protein coupled receptor (GPCR) including improving insulin sensitivity, inhibiting white adipose tissue accumulation and suppressing inflammation [36]. In T2D patients, reduced abundance of SCFAs-producing bacteria results in decreased SCFAs levels, which may promote the development of IR and T2D [37]. However, clinical and animal studies have suggested that fecal SCFAs levels are positively associated with body weight and IR [38]. The role of SCFAs in IR and T2D is thus controversial and needs further investigation. 

#### 1.1.3. Primary and Secondary Bile Acid Metabolites

Bile acids (BAs) are primarily synthesized from cholesterols in hepatocytes. Primary BAs including cholic acid (CA) and chenodeoxycholic acid (CDCA) are synthesized from classical pathways (by cytochrome P450 family 7 superfamily A polypeptide 1, CYP7A1) and alternative pathways (by CYP27A1) in human. The primary BAs are then conjugated to glycine or taurine as taurocholic acid (TCA), glycocholic acid (GCA), taurine chenodeoxycholic acid (TCDCA) and glycine chenodeoxycholic acid (GCDCA). After these bile acids enter intestine especially for the small intestine, the bile salt hydrolase (BSH) of gut microbiota can transform these conjugated BAs into free BAs and secondary BAs including deoxycholic acid (DCA), lithocholic acid (LCA) and UDCA in human. 90–95% BAs in intestine are re-absorbed by apical-sodium-dependent BA transporter (ASBT) in the distal ileum and transported to liver through hepatic portal vein by organic solute transporter alpha/beta (OSTα/β). There is a negative feedback regulatory mechanism for the BA synthesis. When BAs enter intestine, they activate farnesoid X receptor (FXR) to upregulate the expression of fibroblast growth factor 15 (FGF15) in mice or FGF19 in humans. FGF15/FGF19 bind to fibroblast growth factor receptor 4 (FGFR4) to inhibit the expression of CYP7A1, thereby inhibiting the synthesis of primary BAs. BA metabolism is also regulated by small heterodimer partner (SHP) since SHP mediates downregulation of CYP7A1, thereby inhibiting the BA synthesis at an inhibitory feedback manner. BAs act as anti-microbial agents in gut; only specific bacterial populations that can endure high BA concentration can survive well in the gut [39,40]. 

Since IR and T2D are associated with dysregulated BA metabolism in both animal and human studies, the impact of gut microbiota on BA metabolism is of great interest. Even though total BA levels in T2D subjects are increased, the changes of primary and secondary bile acids in numerous T2D studies does not show a consistent trend [41]. BA ligands can bind to either cell surface receptors including TGR5 and sphingosine-1-phosphate receptor (S1PR) or nuclear receptors including FXR, vitamin D receptor (VDR), pregnane X receptor (PXR) [42]. TGR5 is expressed in enteroendocrine L-cells, WAT, brown adipose tissue (BAT), skeletal muscle, liver and the brain. Natural TGR5 agonists include LCA, DCA, CDCA and CA [43]. Conjugated BA binds to SIPR2 to activate nuclear sphingosine kinase-2 in hepatocytes via ERK1/2 and Akt signaling pathways [44]. FXR is expressed in liver, intestine, kidneys and white adipose tissue (WAT). Natural FXR agonists are CDCA, DCA, CA and LCA by potency order, while T-alpha, T-beta MCA and UDCA are potential antagonists [45]. BA regulates glucose homeostasis by directly acting on FXR and TGR5 in the intestine, liver, and pancreas. In the intestine, FXR-deficient mice exhibit delayed kinetics of glucose absorption. FXR inhibits hepatic glycolysis and reduces post-prandial glucose utilization, whereas FGF15/19 increases glycogenesis [46]. FXR and TGR5 are both expressed in pancreatic β-cells and stimulate glucagon synthesis and glucose-induced insulin secretion [47]. VDR is activated by LCA, which stimulates the expression of CYP3A and activates ERK 1/2 pathway to inhibit the insulin signaling pathway [48,49]. PXR can be activated by CDCA, DCA and CA and suppressed by T-α- and T-β-MCA [50]. PXR activation has been shown to impair glucose tolerance and downregulate genes controlling gluconeogenesis [51,52]. 

#### 1.1.4. Protein and Peptide Metabolites

Branched-chain amino acids (BCAA) products

BCAA (leucine, isoleucine, and valine) are essential nutrients with important roles in protein synthesis. Gut microbiota is the main source of circulating BCAA through biosynthesis and modification of absorption. Elevated circulating BCAA has been related to metabolic disorders like IR and T2D [53]. Firstly, IR causes dysbiosis of gut microbiota, which promotes the gut environment to change from saccharolytic fermentation-dominant to proteolytic fermentation dominant, resulting in increased harmful metabolites derived from BCAA. Meanwhile, the structure profile of gut microbiota is influenced by nutrients (amino acids like BCAA) and environmental factors (local GI pH) that are dysregulated in IR. Secondly, IR and T2D are associated with a reduction in BCAA catabolism in peripheral tissues, thus affecting insulin, glucagon and GLP-1 secretion [54]. The role of branched-chain fatty acids (BCFA), as the bacterial metabolites of BCAA, in regulating glucose metabolism is not well studied. Studies show that BCFA inhibits both lipolysis and lipogenesis in human adipocytes, and isobutyric acid potentiates insulin-stimulated glucose uptake in rat adipocytes, suggesting BCFA affects glucose metabolism in adipocytes and may contribute to the development of IR and T2D [55]. 

Aromatic amino acids (AAA) products

Indoles: Indoles and their derivatives are the intermediate metabolites of the tryptophan metabolism pathways. By direct transformation, tryptophan is processed into tryptamine, indole-3-aldehyde (IAld), indole-3-acid-acetic (IAA), indole-3-propionic acid (IPA), and indole acrylic acid (IA) by gut microbiota via different metabolic routes. Several indole derivatives may be associated with the development of the metabolic syndromes. Indole regulates the insulin secretion and sensitivity by manipulating GLP-1 production in enteroendocrine L cells. The activating mechanism involves the rapid inhibition of voltage-gated K^+^ channels stimulating GLP-1 secretion, whereas long-term exposure to indole inhibits ATP synthesis to reduce GLP-1 secretion [56]. Indole is also metabolized in the liver into indoxyl sulfate that is one of the contributing factors of kidney failure in T2D [57]. 

(Poly) amines and other amino acids products

Amines are primarily derived from bacterial fermentation of amino acids in the gut, including phenylethylamine (phenylalanine), tryptamine (tryptophan), tyramine (tyrosine), agmatine (arginine), histamine (histidine) and cadaverine (lysine). Studies have shown increased serum levels of putrescine and spermine in T2D, whereas the functions of polyamines on glucose metabolism have not been systemically studied yet [58]. Moreover, imidazole propionate (histidine products) is elevated in T2D and impairs glucose tolerance in vivo [59]. At the cellular level, it inhibits insulin signaling through activation of p38γ/p62/mTORC1 signaling cascade. 

#### 1.1.5. Lecithin, Choline and L-Carnitine Products: TMA and TMAO

Trimethylamine (TMA) is produced from dietary choline, phosphatidylcholine, and carnitine exclusively by microbiota in the gut. TMA produced by gut microbiota is absorbed in the bloodstream and metabolized in the liver into trimethylamine N-oxide (TMAO). TMAO levels in plasma are positively associated with increased risk of IR and T2D [60]. The deletion of flavin monooxygenase (FMO3), which is the catalyzed enzyme converting TMA to TMAO, protects mice from obesity and IR [61]. The reduction of TMAO by the dietary changes has been shown to be associated with improved insulin sensitivity in T2D [62]. 

#### 1.1.6. Gases Products: Methane and Hydrogen Sulfide 

Gases products including methane and hydrogen sulfide have been shown to regulate metabolic function and are involved in the treatment and development of T2D. Methane is produced from carbohydrates by metabolic actions of methane producers (methanogens) in gut [63]. Studies have revealed that methanogens and methane are significantly increased in HFD-fed mice and positively correlated with GLP-1 secretion. Methane enhanced cAMP level and stimulated GLP-1 secretion in L-cells [64]. Methane also facilitates SCFA production by consumption of dihydrogen and carbon dioxide [65]. Reduced methane level has been found in insulin resistance, indicating that methane alterations directly alters GLP-1 secretion in type 2 diabetes [66]. However, the impact of methane on type 2 diabetes has not been confirmed yet since changes of methane level are not consistent, whilst excessive methane causes GI discomfort and prolong GI transit. 

Hydrogen sulfide (H_2_S) has been widely studied in the regulation of glucose metabolism homeostasis. Besides endogenous production by pancreatic beta cells and insulin sensitive tissues (liver, adipose and skeletal muscles), H_2_S can be produced by gut microbiota. H_2_S level in the blood are reduced in diabetic patients [67]. H_2_S exhibits multiple regulatory roles in insulin sensitivity and insulin secretion. In pancreatic beta cells, H_2_S inhibits insulin secretion via activating K^+^ channels [68]. In the liver, H_2_S inhibits glucose uptake and stimulates glycogenolysis [69]. In contrast to H_2_S function in the liver, reports of H_2_S functions in adipose tissue are controversial, while both stimulatory and inhibitory effects of H_2_S on glucose uptake in adipose tissues have been reported [70]. In skeletal muscle, H_2_S exhibits beneficial effects on insulin sensitivity and enhances glucose uptake [71]. Overall, excessive H_2_S production may contribute to type 2 diabetes and the underlying molecular mechanisms still require further investigation. 

### 1.2. Treatment of T2D by Probiotics 

#### 1.2.1. Probiotics Interventions in Animal Models of Diabetes

Studies have shown that probiotics exhibit beneficial effects on IR in animal models of diabetes (Table 2). The biological effects of probiotics including *Lactobacillus* spp. and *Bifidobacterium* spp. on glucose intolerance and IR have been extensively investigated in diabetic animal models. For example, administration of *Lactobacillus plantarum* CCFM0236 was found to ameliorate insulin resistance, systemic inflammation and pancreas β-cell dysfunction in high fat diet (HFD) and streptomycin (STZ)-induced diabetic mice [72]. *Lactobacillus plantarum* Ln4 reduced weight gain and alleviated insulin resistance by improving oral glucose tolerance test (OGTT), insulin tolerance test (ITT) and homeostatic model assessment for insulin resistance (HOMA-IR) indexes in mice fed on HFD [73]. *Lactobacillus fermentum* MTCC 5689 treatment has been shown to improve insulin resistance and prevent the development of diabetes in HFD-induced diabetic mice [74]. Moreover, administration of *Lactobacillus paracasei* TD062 improved the glucose homeostasis and enhanced insulin signaling pathway, preventing the development of T2D [75]. A multiple probiotics formula including *Lactobacillus reuteri*, *L. crispatus*, and *Bacillus subtilis* has been investigated in STZ-induced diabetic rats, revealing daily consumption of probiotics formula is effective in alleviating the glucose intolerance and the impaired insulin secretion [76]. Another composite probiotic including 10 *Lactobacillus* strains and four yeast strains were found to alleviate T2D in *db/db* mice by reducing fasting blood glucose (FBG), OGTT and HbA1c indexes and enhancing GLP-1 secretion [77]. Nano-selenium-enriched *Bifidobacterium* longum has been shown to delay the onset of STZ-induced diabetes and ameliorate the high glucose-induced renal function damage [78]. *B. longum* DD98 and selenium-enriched *B. longum* DD98 reduced the levels of FBG and HbA1c and improved the glucose tolerance in HFD and STZ-induced diabetic mice [79]. Moreover, inactivated *B. longum* BR-108 has been reported to reduce blood glucose level in a Tsumura Suzuki Obese Diabetes (TSOD) mouse model of diabetes [80]. *B. animalis* 01 treatment improved OGTT and HOMA-IR indexes and suppressed pro-inflammatory cytokines in HFD and STZ-induced diabetic rats [81]. 

#### 1.2.2. Probiotics Randomized Control Trial (RCT) Interventions Studies in Human with T2D

We summarized the probiotics studies including single-strain probiotic interventions and multi-strain probiotic interventions in T2D patients (Table 3). These studies demonstrated the regulatory effects of probiotics on the management of blood glucose level, HbA1c and body weight, which could be beneficial for restoring glucose homeostasis in T2D. 

**i.** 
**Single-strain probiotic intervention in T2D patients**


One study investigated the metabolic effects of *Lactobacillus reuteri* DSM 17938 in T2D patients [93]. Briefly, 46 T2D patients took placebo or 10^8^–10^10^ CFU/d of *L. reuteri* DSM 17938 for 12 weeks. The results showed that this probiotic did not affect HbA1c level in participants. However, participants who received *L. reuteri* DSM 17938 exhibited the increase in insulin sensitivity index (ISI). The effects of *Lactobacillus casei* 431^®^ was investigated in Iranian adults with T2D [94]. Subjects in the probiotic group (*n* = 20) consumed at least 10^8^ CFU/day of *Lactobacillus casei* 431^®^ for 8 weeks while the control group (*n* = 20) consumed placebo. The results showed FBG, insulin level, insulin resistance and fetuin-A level significantly reduced while the level of SIRT1, a key anti-aging protein and regulator of insulin sensitivity, increased in the probiotic-treated group. 

**ii.** 
**Multi-strain probiotic intervention on T2D patients**


A 24-week clinical trial to assess the effects of probiotics was conducted on prediabetic subjects. Briefly, 120 participants were given either containing *Lactobacillus acidophilus*, *Bifidobacterium lactis*, *Bifidobacterium bifidum*, and *Bifi-dobacterium longum*, or synbiotic comprising the mentioned probiotics with an insulin-based prebiotic, or placebo [95]. Compared with the placebo, symbiotic and probiotics supplementation reduce FPG, FIL and HOMA-IR levels, indicative of improved glycemic indices in prediabetic subjects. An RCT intervention in 78 Saudi T2D patients was performed to characterize the beneficial effects of probiotics [96]. After multi-strain probiotics supplementation for 12 weeks, participants in the probiotics group showed an improvement in WHT and HOMA-IR. The effects of multi-strain probiotics (14 live probiotic strains of *Lactobacillus*, *Lactococcus*, *Bifidobacterium*, *Propionibacterium* and *Acetobacter*) vs. placebo on insulin resistance were studied in 53 T2D patients [97]. The patients were randomly assigned to receive probiotics or placebo for 8 weeks. Supplementation with probiotics for 8 weeks significantly reduced HOMA-IR. In probiotics responders, a significant reduction of HbA1c was found when compared with non-responders. Moreover, pro-inflammatory markers including TNF-alpha and IL-1beta were also significantly reduced in the probiotics-treated group. 

#### 1.2.3. Molecular Mechanism of Probiotics Intervention on T2D

Studies have shown that probiotics can ameliorate IR, pancreatic β-cell dysfunction and hyperglycemia [98], whereas limited studies have evaluated the molecular mechanisms of probiotics intervention in T2D. Mechanistically, probiotics alleviate T2D-associated pathologies by repairing intestinal barrier, suppressing inflammatory responses, reducing oxidative stress, restoring energy metabolism, and producing beneficial microbial metabolites including SCFA and BA (Figure 3). Specifically, one study showed that *Lactobacillus acidophilus* KLDS1.0901 improved intestinal barrier function, and suppressed inflammatory responses in liver and colon in an animal model of diabetes [99]. Another study showed *Lactobacillus casei* CCFM419 enhanced SCFA and GLP-1 production and reduced the levels of pro-inflammatory markers in diabetic mice [83]. *Akkermansia muciniphila* treatment has been shown to improve liver function, alleviate oxidative stress and suppress inflammation in diabetic rats [100]. *Lactobacillus casei* was found to enhance SCFA production as well as GLP-1 and PYY secretion in diabetic mice [101]. *Lactobacillus reuteri* DSM 17938 supplementation was found to enhance gut-microbial diversity and serum DCA levels [93]. 

## 2. Discussion 

### 2.1. Diagnostic Applications in the Clinic

Currently, gut microbiome-based metabolomics studies are still in the early stage of T2D. Gut-microbial biomarkers derived by metabolomics approaches may provide valuable insights into the development of IR, as well as the traditional risk factors of T2D. One advantage of profiling metabolites rather than the microbiota per se is that it overcomes the pitfall arises from functional redundancy and focuses on the functions of the microbiome. The biomarkers derived from prospective studies for predicting the risk of pre-diabetes and diabetes may be applicable to the prevention of metabolic syndrome and diabetes. Furthermore, the metabolomics approaches can be helpful for the diagnosis and treatment of T2D, which provides a personalized treatment strategy for predicting the development and appropriate medication of T2D. 

### 2.2. Drug Discovery Based on Gut Microbiota-Derived Metabolites

With the development of high-throughput sequencing of gut microbiota, scientists and pharmacological companies are mining small-molecule drug discovery programs using conventional drug discovery and novel synthetic biology approaches. Functional metagenomics have helped investigators identify bioactive molecules and targets, followed by the identification of homologous gene families [102]. An important family of G protein-coupled receptor (GPCR) ligands namely *N*-acyl amides produced by gut microbiota is shown to be agonists of receptors that have important functions for gastrointestinal and metabolic diseases, such as the endocannabinoid receptor GPR119 [103]. Another study used a novel approach combining computational and synthetic biology and characterized a series of microbiota-derived metabolites that can inhibit host proteases [104]. Moreover, a “chemistry-forward” approach of screening of GPCR ligands from gut microbiota metabolomes has revealed gut microbes produce ligands for many GPCRs and the microbiota-derived GPCR ligands have a profound impact on host physiology [105]. Such approaches used for mining gut microbiota-derived metabolites and novel compounds provide a potential strategy for discovering drugs to treat T2D.

### 2.3. Alternative Therapeutic Options

#### 2.3.1. Gut Microbiota-Derived Probiotics (GPs)

Current probiotic supplements recommended for diabetic patients are mainly *Bifidobacterium* spp., *Lactobacillus* spp., and yeasts, which are culturable, aerotolerant and can be produced in an industrial scale [106]. By contrast, the novel probiotics for T2D include important gut bacteria in human gut which are reduced in T2D. However, it is difficult to culture these gut bacteria that are extremely sensitive to oxygen, which presents a great challenge in terms of isolation, cultivation and industrial production and formulation. Unlike the common probiotics, the GPs directly from human may also require stricter evaluation procedures in terms of safety and efficacy, which may need new drug approval procedures according to the FDA. 

#### 2.3.2. Prebiotics Supplement

Besides of probiotics, modulation of gut microbiota can also be achieved using prebiotics. Prebiotics is a mixture of nondigestible food ingredients that promote the growth of beneficial microbes and suppress growth of pathogenic microbes in GI tract [107]. It brings numerous benefits to the host including normalization of GI pH value, modulation of immune system, reduction of hyperlipidemia and improvement of cation ions absorption [108]. Several mechanisms of prebiotics action on gut microbiota and host have been identified so far. First, the production of beneficial microbial metabolites (SCFA) by beneficial microbes, such as *Bifidobacterium* and *Lactobacillus,* is promoted by prebiotics [109]. Second, prebiotics supplement suppresses endotoxin level by inhibiting the growth and colonization of harmful bacteria [110]. Third, prebiotics improve cation ions absorption possibly by regulating pH value in GI tract [111]. Prebiotics can be used as a supplement or additional support for probiotics. The complementary symbiotic comprising probiotics plus prebiotics can be more effective than probiotics formulation alone in promoting human health. 

#### 2.3.3. Fecal Microbiota Transplantation (FMT)

FMT is an interesting approach to modulate gut microbiota and has been used to correct gut microbiota dysbiosis in clinical trials. FMT from lean donors has been implanted to obese subjects, after which metabolic syndrome and insulin sensitivity were improved by FMT, suggesting modulation of gut microbiome could be considered as a novel therapeutic target for the treatment of IR [112]. The reasons for the beneficial properties of gut microbiota could be attributed to its enhancement of levels of gut microbial metabolites levels including SCFAs and BAs [113]. A pilot FMT study in nine obese individuals was conducted using fecal samples from lean healthy donors [114]. The recipients displayed a significant improvement in insulin sensitivity and the beneficial effects were confirmed by a larger-scale follow-up study, in which the recipients showed a reduction of HbA1c at 6 weeks. However, the insulin sensitivity and the composition of gut microbiota switched back to baseline after 18 weeks post-intervention. Studies also found that FMT treatment does not show beneficial effects on subjects with severe IR, suggesting manipulating gut microbiota may only help maintain the glucose level and insulin sensitivity in the early stage of T2D [112]. Nevertheless, FMT is a potential personalized approach for alleviating glucose intolerance and IR in metabolic syndrome and T2D. 

## 3. Conclusions

Both animal and human data provide strong evidence of both beneficial and harmful roles of the microbial metabolites in the prevention and development of IR and T2D. Numerous microbial metabolites, such as LPS, SCFAs, BA, TMAO, are correlated with the development of IR and T2D in humans. However, the metabolic consequences of changes in other microbial metabolites in diabetes are not fully understood. For example, the biological functions of succinate derived from saccharolytic fermentation and BCFA, phenolic and indole compounds derived from proteolytic fermentation in T2D have not been well studied yet. Further studies on this area may provide new understandings of gut microbes and novel strategies for preventing and treating IR and T2D. For the current therapeutics of T2D, probiotics are reported to have the beneficial properties of attenuating IR, but the results are not consistent. Further investigation using standardized probiotics in combination with prebiotic and anti-diabetic medications may provide more useful information regarding the efficacy of probiotics on T2D. 

## Figures and Tables

**Figure 1 ijms-22-12846-f001:**
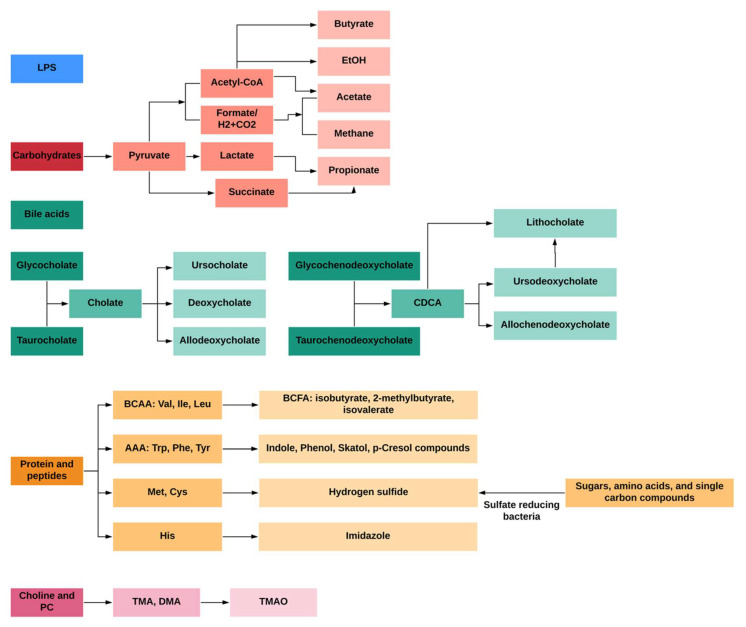
General profiles of gut-microbial metabolites from different dietary and endogenous components in humans.

**Figure 2 ijms-22-12846-f002:**
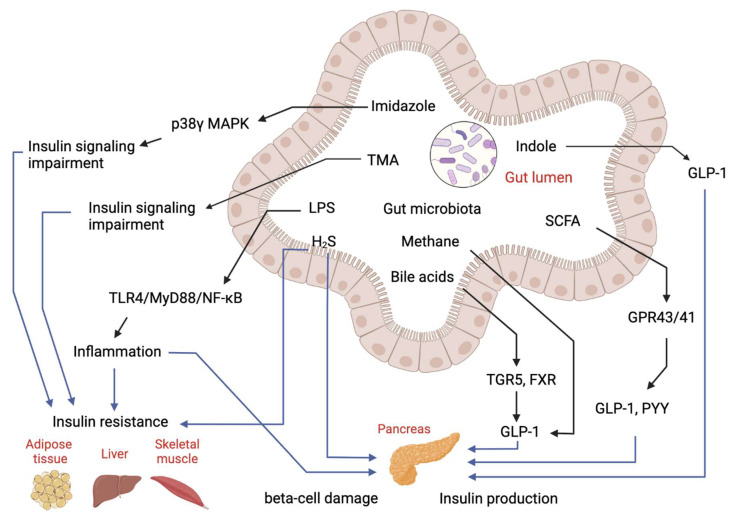
**Regulatory effects of gut-microbial metabolites on insulin sensitivity and insulin production.** Imidazole, TMA, LPS and hydrogen sulfide (H_2_S) can cause either insulin resistance or beta-cell damage to impair glucose homeostasis. Bile acids, SCFA and indole can stimulate GLP-1 production to manipulate insulin production and secretion to regulate glucose level.

**Figure 3 ijms-22-12846-f003:**
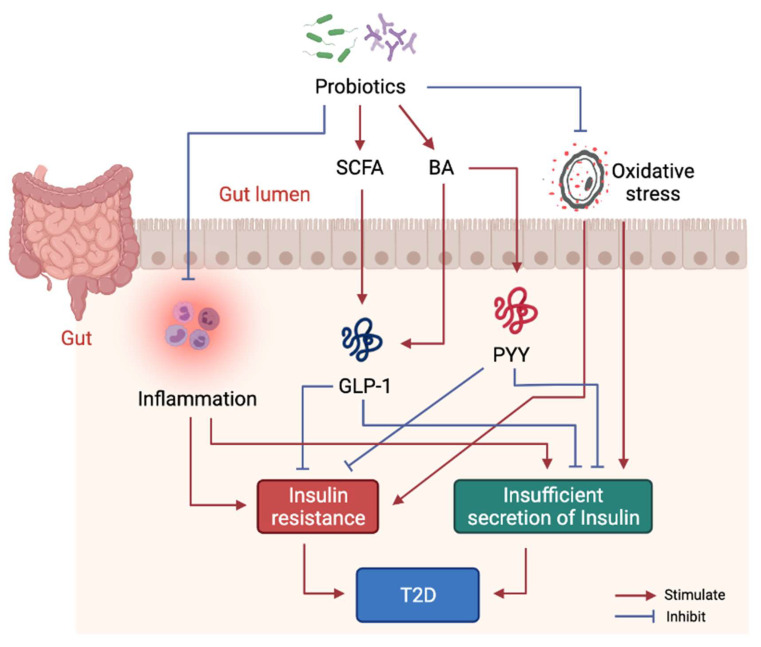
Molecular mechanism of probiotics intervention on T2D. Probiotics in gut help produce beneficial metabolites including SCFA and some BA to stimulate GLP-1 and PYY secretion, thus, to alleviate insulin resistance and dysfunction of insulin secretion. Probiotics also suppress systemic inflammation by modulating gut microbiota structure.

**Table 1 ijms-22-12846-t001:** Concentrations of gut microbial metabolites in human healthy host.

Category	Metabolite	Serum/Plasma	Urine	Feces/Colon	References
LPS	LPS	0.39 ± 0.06 EU/mL	–	0.27 ± 0.04 EU/mL in feces	[10]
SCFAs	Acetate	5–200 µM	82.89 ± 60.0 μM	35.86 ± 16.8 µmol/g in feces	[11]
SCFAs	Butyrate	<12 µM	2.98 ± 1.88 μM	6.35 ± 3.13 µmol g in feces	
SCFAs	Propionate	<13 µM	108.2 ± 78.1 μM	11.40 ± 4.74 µmol g in feces	
SCFAs	Succinate	5–200 µM	10 ± 0.2 μM	3.1 + 0.9 mmol/kg in the proximal colon; 2.1 ± 1.0 mmol/kg in the sigmoid colon	[7]
BCFA	Total	–	–	18.87 mmol/kg	[7]
BCFA	isobutyrate	2.6–4.7 µM	–	0.04–0.24 mg/g	[12]
BCFA	2-methylbutyrate	–	–	4079.7 nmol/g wet feces	
BCFA	isovalerate	11.2–44.4 µM	–	0.05–0.37 mg/g	
Amino acids	Total amines (agmatine, cadaverine, histamine, phenylethylamine, putrescine, spermidine, spermine, tryptamine and tyramine)	–	–	22.32 mmol/kg	[7]
Amino acids	Ammonia	22–55 µM	–	160.93 mmol/kg	[7]
Amino acids	Phenolic acids	–	–	2.39 mmol/kg (total phenols)	[13]
Amino acids	Indole	–	–	2.6 mM	[14]
Amino acids	*p*–cresol	5.556 ± 9.259 µM	52.6 (38.8–71.0) umol/mmol creatinine	2.12 mmol/kg	[15]
Bile acids	Deoxycholic acid	0.57 ± 0.35 µM	–	1920.10 +/− 1390.50 nmol/g dry feces	[16]
Bile acids	Lithocholic acid	0.0103 µM	–	1016.60 +/− 647.31 nmol/g dry feces	
Bile acids	Ursodesoxycholic acid	0.1975 µM	–	27.05 +/− 61.13 nmol/g dry feces	[16]
Choline	TMA	26.55 (7.07) µM	0.24–2.33 µmol/mmol creatinine	–	[17]
Choline	TMAO	38.81 ± 20.37 µM	20–125 µmol/mmol creatinine	18417.506 (9541.599–27,293.412) nmol/g wet feces	[18]
Gas	Methane	–	–	–	
Gas	Carbon dioxide	–	–	–	
Gas	Hydrogen sulfide	37.6 (27.4–41.3) µM	–	–	[19]

**Table 2 ijms-22-12846-t002:** Probiotics intervention in animal model of diabetes.

Probiotic Species/Strains	Disease Model	Main Results	References
*Lactobacillus plantarum* CCFM0236	HFD+STZ	Blood glucose ↓, leptin level ↓, insulin resistance ↓	[72]
*Lactobacillus plantarum* Ln4	HFD	Insulin resistance ↓, insulin response ↑	[73]
*Lactobacillus fermentum* MTCC 5689, *Lactobacillus plantarum* MTCC 5690	HFD	Glucose ↓, HbA1c↓, plasma insulin ↓, HOMA-IR ↓	[74]
*Lactobacillus paracasei* TD062	HFD+STZ	FBG↓, Glucose tolerance ↓	[75]
*Lactobacillus reuteri, Lactobacillus crispatus* and *Bacillus subtiliso*	STZ	Plasma glucose ↓, HbA1c ↓, plasma insulin ↑	[76]
*Lactobacillus kefiranofaciens*, *Lactobacillus plantarum*, *Lactobacillus helveticus*, *Lactococcus lactis* and *Issatchenkia orientalis*	*db/db*	FBG ↓, OGTT ↓, HbAlc ↓ IRI ↓, plasma TC ↓, TG ↓, LDL-C ↓,	[77]
Nano-selenium-enriched *Bifidobacterium* longum	STZ	Blood glucose ↓, renal function damage ↓	[78]
*Bifidobacterium longum* DD98 and selenium-enriched *B. longum* DD98	HFD+STZ	FBG and HbA1c ↓	[79]
Inactivated *Bifidobacterium longum* BR-108	TSOD mouse	Blood glucose ↓	[80]
*Bifidobacterium animalis* 01	HFD+STZ	OGTT and HOMA-IR ↓, pro-inflammatory cytokines ↓	[81]
*Lactobacillus plantarum* OLL2712	HFD	Blood glucose ↓, IL-1beta ↓	[82]
*Lactobacillus casei* CCFM419	HFD+STZ	FBG ↓, glucose intolerance↓, insulin resistance ↓, TNF-alpha and IL-6 ↓, GLP-1 ↑	[83]
*Lactobacillus rhamnosus* NCDC 17	HFD+STZ	FBG ↓, plasma insulin ↓, HbA1c ↓, free fatty acids ↓, TG ↓ and TC ↓,	[84]
*Lactobacillus paracasei* NL41	HFD+STZ	Insulin resistance↓, HbA1c ↓, glucagon ↓ and leptin ↓, oxidative stress status ↓	[85]
*Lactobacillus rhamnosus, Lactobacillus acidophilus* and *Bifidobacterium bifidum*	HFD	Plasma glucose ↓, intestinal permeability ↓, LPS translocation ↓, systemic low-grade inflammation ↓	[86]
*Clostridium butyricum* CGMCC0313.1	*Db/db* mice and HFD+STZ	FBG ↓, HbA1c ↓, GLP-1 ↑ and inflammatory responses ↓	[87]
*Lactobacillus salivarius* AP-32 and *L. reuteri* GL-104	*db/db* mice	FBG ↓, TG ↓, TC ↓	[88]
*Lactobacillus plantarum* HAC01	HFD+STZ	FBG ↓, HbA1c ↓ and insulin-positive β-cell mass ↑	[89]
*Lactobacillus delbrueckii* subsp. lactis PTCC1057	STZ	FBG ↓, fetuin-A ↓ and sestrin ↑	[90]
*Streptococcus thermophilus*	Zucker diabetic fatty (ZDF)	FBG ↓, glucose intolerance ↓, TC ↓, LPS ↓, IL-6 ↓, TNF-α ↓ and IL-10 ↑	[91]
*Lactobacillus plantarum*, *L. bulgaricus*, *L casei*, *L. acidophilus*, *Bifidobacterium infantis*, *B. longum*, *B. breve*	HFD+STZ	Plasma glucose ↓, GLP-1 ↑ and total antioxidant capacity ↑	[92]

**Table 3 ijms-22-12846-t003:** Probiotics RCT intervention in patients with T2D.

Probiotic Species/Strains	Period	Sample Size	Main Results	References
*Lactobacillus reuteri* DSM 17938	12 weeks	46 Placebo (*n* = 15) *L. reuteri* low (*n* = 15) *L. reuteri* high (*n* = 14)	Insulin sensitivity index (ISI) ↑, HbA1c not affected	[93]
*Lactobacillus case* 431^®^	8 weeks	40 Probiotic (*n* = 20) and placebo (*n* = 20)	FBG ↓, insulin ↓ and insulin resistance ↓	[94]
*Lactobacillus acidophilus, Bifidobacterium lactis, B.bifidum, B. longum*	24 weeks	85 (27 in probiotic, 30 in synbiotic and 28 in placebo groups)	FPG ↓, HbA1c ↓and HOMA-IR ↓	[95]
*Bifidobacterium bifidum* W23, *Bifidobacterium lactis* W52, *Lactobacillus acidophilus* W37, *Lactobacillus brevis* W63, *Lactobacillus casei* W56, *Lactobacillus salivarius* W24, *Lactococcus lactis* W19 and *Lactococcus lactis* W58	12 weeks	78 placebo (*n* = 39) and probiotics (*n* = 39).	HOMA-IR ↓	[96]
Symbiter, containing 14 alive probiotic strains of *Lactobacillus*, *Lactococcus*, *Bifidobacterium*, *Propionibacterium* and *Acetobacter* genera.	8 weeks	53 probiotic (*n* = 31) and placebo (*n* = 22)	HOMA-IR ↓, HbA1c ↓, TNF-α ↓ and IL-1β ↓	[97]

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
