# Peer review of "Gut-Microbial Metabolites, Probiotics and Their Roles in Type 2 Diabetes"

_ijms, 2021, doi:10.3390/ijms222312846_

Round 1

Reviewer 1 Report

Major comments:

  1. Edits to Figure 1 to include additional sources for hydrogen sulfide production via sulfate reducing bacteria from the gut microbiome. As many previous publications have reported that the metabolites for hydrogen sulfide production (via sulfate reducing bacteria) utilize sugars, amino acids, and single carbon compounds (ie methanol, carbon monoxide).
  2. In this review on microbiome metabolites on development of diabetes, it is important to include the prior animal studies on methane and hydrogen sulfide impact in prevention/development of diabetes. These gases are mentioned in Section 1.1 but no further elaboration/explanation are provided as to the role of these gases. This is a major oversight by the authors given the number of studies showing impact of methane/hydrogen sulfide on the pancreatic cells and insulin resistance.
  3. Section 1.2 is focused on treatment of T2D by probiotics, which is an important topic to cover. However, section 1.1 focuses on metabolites of development of IR and T2D. The 2 sections appear disjointed for this review paper. There is no review provided in section 1.1 as to the role of probioitics in development of IR/T2D. At the same time, section 1.2 focuses on treatments by probiotics, but does not mention treatments by metabolites (from section 1.1). These are 2 separate topics and not enough review/discussion is provided to either section to make this comprehensive. Would strongly suggest including additional information on probiotics into section 1.1, and treatment via metabolites for section 1.2 (as there are animal studies available regarding treatment).
  4. Section 2 (discussion) appears to focus more on future direction rather than mechanistic explanation of discussed metabolites/probioitics causing IR/T2D. Section 2.2 does provide a discussion regarding metagenomics which is of great benefit for this paper. But none of the other pathways that are detailed in Section 1 are mentioned in the discussion section making this a disjointed section from the rest of the paper.
  5. Section 2.3 would also consider discussion of the impact of prebiotics as a therapeutic option in addition to probiotics for treatment of diabetes/metabolic syndrome.
  6. Section 2.3.2 did not discuss the role of FMT in animal studies. There is a robust amount of data available on the role of FMT for treatment of IR/T2D in animal studies. This review paper should specifically include these studies for comprehensive analysis of the research published.

Minor comments:

  1. Line 44-45, quoted definition of probiotics, no citation is provided.
  2. Line 158-159, literature describes bidirectional influence of metabolic syndrome/IR and dysbiosis, should be important to mention that here as well.
  3. Line 281-282 should have a citation for the sentence.

Author Response

Dear Reviewer, 

Thank you very much for your valuable comments. We sincerely appreciate your efforts toward this manuscript. We have revised the manuscript following reviewer’s kind suggestions and added some references.

Major comments:

  1. Edits to Figure 1 to include additional sources for hydrogen sulfide production via sulfate reducing bacteria from the gut microbiome. As many previous publications have reported that the metabolites for hydrogen sulfide production (via sulfate reducing bacteria) utilize sugars, amino acids, and single carbon compounds (ie methanol, carbon monoxide).

Answer: We have added additional sources for hydrogen sulfide production via sulfate reducing bacteria from the gut microbiome in Figure.1.

  1. In this review on microbiome metabolites on development of diabetes, it is important to include the prior animal studies on methane and hydrogen sulfide impact in prevention/development of diabetes. These gases are mentioned in Section 1.1 but no further elaboration/explanation are provided as to the role of these gases. This is a major oversight by the authors given the number of studies showing impact of methane/hydrogen sulfide on the pancreatic cells and insulin resistance.

Answer: Thank you for your important suggestions. We have added review content of biological actions of methane and hydrogen sulfide on pancreatic cells and insulin resistance. We have also revised Figure.2 by including the molecular pathways of gases action in type 2 diabetes.

  1. Section 1.2 is focused on treatment of T2D by probiotics, which is an important topic to cover. However, section 1.1 focuses on metabolites of development of IR and T2D. The 2 sections appear disjointed for this review paper. There is no review provided in section 1.1 as to the role of probiotics in development of IR/T2D. At the same time, section 1.2 focuses on treatments by probiotics, but does not mention treatments by metabolites (from section 1.1). These are 2 separate topics and not enough review/discussion is provided to either section to make this comprehensive. Would strongly suggest including additional information on probiotics into section 1.1, and treatment via metabolites for section 1.2 (as there are animal studies available regarding treatment).

Answer: It is an excellent advice. We have made several revisions towards these suggestions. In section 1.1, we have discussed the detrimental and beneficial effects of microbial metabolites on diabetes. These metabolites are served as either pathological or therapeutic role in pancreatic cells dysfunction and insulin resistance. We also have discussed that the actions of probiotics intervention on T2D are mainly dependent on SCFA and bile acids in section 1.2.

  1. Section 2 (discussion) appears to focus more on future direction rather than mechanistic explanation of discussed metabolites/probioitics causing IR/T2D. Section 2.2 does provide a discussion regarding metagenomics which is of great benefit for this paper. But none of the other pathways that are detailed in Section 1 are mentioned in the discussion section making this a disjointed section from the rest of the paper.

Answer: In section 2.1 and 2.2, we have summarised two application directions for the microbial metabolites in T2D. These T2D-associated metabolites can be developed for either diagnostic or therapeutic use. For the metabolic pathways shown in section 1, we have included regulatory effects of gut-microbial metabolites on insulin sensitivity and insulin production in revised figure.2.

  1. Section 2.3 would also consider discussion of the impact of prebiotics as a therapeutic option in addition to probiotics for treatment of diabetes/metabolic syndrome.

Answer: We have added a short discussion to summarize the biological functions and applications of prebiotics on human health.

  1. Section 2.3.2 did not discuss the role of FMT in animal studies. There is a robust amount of data available on the role of FMT for treatment of IR/T2D in animal studies. This review paper should specifically include these studies for comprehensive analysis of the research published.

Answer: In section 2.3.2, we mainly discussed the importance of FMT as a potential treatment of IR/T2D. Since our manuscript primarily focuses on microbial metabolites and probiotics and their potential roles in T2D, the research progress of FMT studies may be out of scope. The detailed FMT research for treatment of IR/T2D including clinical studies and animal studies will be further explored in the future.

Minor comments:

  1. Line 44-45, quoted definition of probiotics, no citation is provided. Cited
  2. Line 158-159, literature describes bidirectional influence of metabolic syndrome/IR and dysbiosis, should be important to mention that here as well. Added
  3. Line 281-282 should have a citation for the sentence. Cited

Answer: We are sorry for the mistake. We have either cited or added the content for these points accordingly in the revised manuscript.

Reviewer 2 Report

Zhai et al., nicely drafted the review on Gut-microbial metabolites, probiotics and their roles 2 in type 2 diabetes. Overall, authors covered the most prominent metabolites and probiotics in  the management of diabetes.

Minor comments:

Figure1: Please modify the figure. Focus on the location.

Table1: Is this human healthy host or human disease?

Legend of Figure 1 & 2 mentioned as Figure 1 & 1. Correct it.

Page 5, Line 5: Elaborate about figure 2 in the text.

Author Response

Thank you very much for your encouraging comments. Please see our replies below:

Figure1: Please modify the figure. Focus on the location.

Answer: We are sorry that we don't really understand what we should correct. Would you please further advise us?

Table1: Is this human healthy host or human disease?

Answer: This is human healthy host. 

Legend of Figure 1 & 2 mentioned as Figure 1 & 1. Correct it.

Answer: We are sorry for the typo. We have corrected it accordingly.

Page 5, Line 5: Elaborate about figure 2 in the text.

Answer: We have briefly described figure 2 with figure legend. 

Reviewer 3 Report

I have completed my evaluation of your manuscript. Due the importance of this review, it would be adisable one more in-depth reading and review of the suggestions in the manuscript text, attached. Don't forget to better conceptualize the metabolites in a more complete way.

Author Response

I have completed my evaluation of your manuscript. Due the importance of this review, it would be adisable one more in-depth reading and review of the suggestions in the manuscript text, attached. Don't forget to better conceptualize the metabolites in a more complete way.

Answer: Thank you very much for your encouraging comments. We have corrected the errors accordingly and revised the wordings following your kind suggestions. 

Reviewer 4 Report

In the review article, " Gut microbial metabolites and probiotics....in type 2 diabetes", Zhai et al., have tried to summarize the role of different metabolites available to the human host from gut microbial activity, on t2d. The review can be much more informative than in its present form. It is not well structured, is repetitive and lacks discussion on future of the topic. 

Author Response

Thanks for your kind comments. We have revised the manuscript by including more information about the role of gut microbial metabolites and probiotics in T2D and strengthening the discussion.  

Round 2

Reviewer 1 Report

Major sections are: 1) Introduction, 3) Discussion, 4) Conclusion. No section 2 available.

Line 227, needs a space for "been confirmed"

Author Response

Thank you for your comments. 

We have corrected the the mistake in line 227.

Reviewer 4 Report

The article requires a lot of work before consideration for publication.

There is lack of overall structure and rhythm in the article. The information provided is incomplete. I acknowledge the topic is too broad but important information should be included. For example, for SCFAs: receptor independent effects are not mentioned. In fact, receptor dependent effects leading to IR and T2D through adipose tissue (esp. for FFAR2) are not included. TMAO is most known for cardiac pathology, which is not mentioned. I believe the article should be more comprehensive.

Author Response

Thank you again for your comments. We have made substantial changes to the manuscript, according to yours and other reviewers' comments.